# Fabrication of PEDOT:PSS/Ag_2_Se Nanowires for Polymer-Based Thermoelectric Applications

**DOI:** 10.3390/polym12122932

**Published:** 2020-12-08

**Authors:** Dabin Park, Minsu Kim, Jooheon Kim

**Affiliations:** School of Chemical Engineering & Materials Science, Chung-Ang University, Seoul 06974, Korea; dragoo@naver.com (D.P.); alstn275@gmail.com (M.K.)

**Keywords:** thermoelectric, silver selenide, nanowire, poly(3,4-ethylenedioxythiopene)-poly(4-styrenesulfonate)

## Abstract

Flexible Ag_2_Se NW/PEDOT:PSS thermoelectric composite films with different Ag_2_Se contents (10, 20, 30, 50, 70, and 80 wt.%) are fabricated. The Ag_2_Se nanowires are first fabricated with solution mixing. After that, Ag_2_Se NW/PEDOT:PSS composite film was fabricated using a simple drop-casting method. To evaluate the potential applications of the Ag_2_Se NW/PEDOT:PSS composite, their thermoelectric properties are analyzed according to their Ag_2_Se contents, and strategies for maximizing the thermoelectric power factor are discussed. The maximum room-temperature power factor of composite film (178.59 μW/m·K^2^) is obtained with 80 wt.% Ag_2_Se nanowires. In addition, the composite film shows outstanding durability after 1000 repeat bending cycles. This work provides an important strategy for the fabrication of high-performance flexible thermoelectric composite films, which can be extended to other inorganic/organic composites and will certainly promote their development and thermoelectric applications.

## 1. Introduction

Thermoelectric devices can directly convert thermal energy to electrical energy, and are promising for addressing the future energy crisis [1,2,3,4]. In particular, highly flexible thermoelectric devices for powering portable/wearable electronics could have the potential to continuously convert body temperature to electricity by using the temperature difference between the human body and the environment [5,6,7,8]. The efficiency of a thermoelectric material is characterized by the dimensionless figure of merit, *ZT = S^2^·σ·T/к*, where *S* is the Seebeck coefficient, *σ* is the electrical conductivity, *к* is the thermal conductivity, and *T* is the absolute temperature. As these equations show, low thermal conductivity and high-power factor (*PF = S^2^·σ*) are important to achieve high *ZT*.

Previous research on high-efficiency thermoelectric (TE) devices has mainly focused on inorganic materials [9,10,11,12]. oxides such as CaMnO_3_, and NaCo_2_O_4_ [13,14], and half-Heusler compounds [15,16]. The improvement of thermoelectric properties of materials is achieved mainly via the energy-filtering or phonon-scattering effects of their nanostructures. While the inorganic TE materials present challenges such as high cost, scarcity of elements, poor mechanical flexibility, and high toxicity [17], the conductive polymers are potential alternatives with the advantages of mechanical flexibility, materials abundance, low cost, and inherently low thermal conductivity [18]. Hence, organic polymer TE materials such as polypyrrole (PPy) [19], polyaniline (PANI) [20], and PEDOT:PSS (poly(3,4-ethylenedioxythiopene)-poly(4-styrenesulfonate) [21]) are investigated for thermoelectric applications. For example, Yao et al. [22] prepared a single-walled carbon nanotube (SWCNT)-PANI composite film to achieve an outstanding power factor of ~176 μW/m·K^2^. Meanwhile, other researchers have prepared PPy/graphene/PANI ternary nanocomposites with high thermoelectric properties by combining in-situ polymerization with solution processing [23].

Among the conductive polymers, PEDOT:PSS is a promising TE material due to the high power factor arising from its outstanding electrical conductivity. For example, Kim et al. studied an improvement in the power factor (~33 μW/m·K^2^) and electrical conductivity (~620 S/cm) of PEDOT:PSS when treated with dimethyl sulfoxide (DMSO) [24]. Similarly, Luo et al. studied the DMSO post-treatment of PEDOT:PSS thin films to give an improve electrical conductivity of ~930 S/cm [25]. However, due to their low Seebeck coefficients, organic thermoelectric materials continue to display lower power factors than inorganic materials.

To address these essential problems, many researchers have attempted to manufacture inorganic/organic composite materials to enhance the Seebeck coefficients of polymer-based TE materials by adding inorganic fillers. For example, Song et al. prepared PEDOT:PSS/Te nanorod composite films to provide an approximately 9-fold enhancement in the power factor compared to that of the pristine PEDOT:PSS [26]. Meanwhile, Zhnag et al. prepared a ball-milled PEDOT:PSS/Bi_2_Te_3_ particles composite film that showed an improved Seebeck coefficient compared to that of the pristine PEDOT:PSS [27].

In the present study, a simple solution-mixing and drop casting method for fabricating a flexible thermoelectric generator based on a thin film composed of one-dimensional Ag_2_Se nanowire (NW) and a conductive polymer is described. The fabricated thermoelectric fabric displays a highest power factor of 178.59 μW/m·K^2^ at room temperature. The thermoelectric properties and durability of composite films at different temperatures are also measured. The present authors consider the flexible composite film consisting of the Ag_2_Se NW and PEDOT:PSS to be an essential contribution to the development of inorganic/organic flexible thermoelectric applications.

## 2. Experimental

### 2.1. Materials

PEDOT:PSS (Clevious PH 1000) was purchased from Heraeus Clevios GmbH (Leverkusen, Germany). Selenium (IV) oxide (SeO_2_, 99%), silver(I) nitride (AgNO3, 99.8%), β-cyclodextrin (C_42_H_70_O_35_, 98%), and L(+)-ascorbic acid (C_6_H_8_O_6_, 99%) were purchased from Daejung Chemical & Metals Co. (Seoul, Korea). Ethylene glycol (EG, C_2_H_6_O_2_, 99.8%), ethanol (C_2_H_5_OH, 98%), dimethyl sulfoxide (DMSO, (CH_3_)_2_SO, 99%) were purchased from Sigma Aldrich (St. Louis, MO, USA). All materials were used without further purification.

### 2.2. Preparation of Ag_2_Se NWs

In a round-bottomed flask, 1g of SeO_2_ and β-cyclodextrin were each dispersed in 200 mL DI water. In a separate glass beaker, 2 g of L(+)-ascorbic acid and 200 mL of DI water were mixed. When each solution was completely dissolved, the ascorbic acid solution was poured to the Se precursor solution. After 4 h, a dispersion of Se NW was obtained. This was centrifuged to remove the supernatant, then the final products were vacuum filtered with volumetric water and ethanol. For the Ag_2_Se synthesis, the selected amount of AgNO_3_ was mixed with 5 mL of DI water. The Ag precursor solution and L(+)-ascorbic were then dropped into the as-synthesized Se NWs solution which had been re-dispersed in EG (400 mL). The molar ratio of [L(+)-ascorbic acid]:[AgNO_3_] was 3:1. This mixture was allowed to stand for 4 h. At last, the synthesized composites were washed with DI water.

### 2.3. Fabrication of the Ag_2_Se NW/PEDOT:PSS Composite Films

First, PEDOT:PSS solution are mixed with 5 vol.% of DMSO. This mixture was then sonicated for 30 min. Various contents of Ag_2_Se NWs (0, 10, 20, 30, 50, 70, and 80 wt.%) were then added, and the mixture was sonicated for an additional 3 h. The resulting Ag_2_Se NW/PEDOT:PSS solution was drop-cast, and then dried at 328 K for 24 h.

### 2.4. Characterization

The micromorphology of samples are analyzed with field-emission transmission electron microscopy (FE-TEM, JEM-2100F, ZEISS, Oberkochen, Germany) and Field-emission scanning electron microscopy. In addition, energy-dispersive X-ray spectroscopy (EDS; NORAN system 7, Thermo Scientific, Seoul, Korea) was used to analyze the elemental mapping of composite samples. The X-ray diffraction (XRD; New D8 ADVANCE, Bruker-AXS, (Billerica, MA, USA) at 40 mA and 40 KV using a Cu Kα radiation (0.154056 nm) source and a scan rate of 1°/s in the 2θ range of 5° to 70° was employed to characterize the crystal structure of the composites. X-ray photoelectron spectroscopy (XPS; K-alpha, Thermo Scientific) was used to investigate the binding energy peaks of the composites. For the Seebeck coefficient and electrical conductivity measurements, 18 mm × 18 mm square film samples of the composites were prepared.

The homemade device made up of a pair of thermocouples and voltmeters was used to analyze the Seebeck coefficient S. The typical 4-probe technique was used to evaluate the electrical conductivity. The bending test of the film was performed using a home-made apparatus. All of the above measurements were performed at room temperature. The formula *κ = ρ*·*α*·*C_P_*, (*ρ*, *α*, and *C_P_* are the bulk density, thermal diffusivity, and specific heat of the materials, respectively) was used to determine the thermal conductivity of the composites. A differential scanning calorimeter (DSC, DSC 131 Evo, Setaram Instrumentation, Caluire, France) was used to measure *C_p_*. The xenon flash method was conducted using a Netzsch LFA 447 Nanoflash, (Selb, Germany) instrument to evaluate *α*.

## 3. Results and Discussion

The Ag_2_Se NWs were fabricated through the following simple synthesis. First, the Se NWs are synthesized with a reduction in SeO_2_. While mixing the SeO_2_, β-cyclodextrin, and L(+)-ascorbic acid in DI water. Nucleation of Se_2-_ ions took place at this stage. Se^2-^ ions were reduced to Se, and a solid crystal nucleus is formed. The reaction proceeds according to Equation (1):SeO_2_ + 2C_6_H_8_O_6_ → Se + 2C_6_H_6_O_6_ + 2H_2_O(1)

The solution-based template-oriented synthesis is a simple and easy strategy for generating 1D selenides. In this approach, the template can act as a physical scaffold where different materials are assembled into nanostructures in a similar form to the original template. To confirm the successful synthesis and surface morphology of Se NWs, XRD and FE-SEM were conducted and the results presented in Appendix A.

The as-synthesized Se NWs and Ag precursor solution are then added to EG to generate the Ag_2_Se NWs within a few hours. During this synthesis, two chemical reactions occur. First, Ag^+^ is reduced to Ag according to Equation (2):2Ag^+^ + C_6_H_8_O_6_ → 2Ag + C_6_H_6_O_6_ + 2H^+^(2)

Then, the Se NWs reacts with Ag to form of orthorhombic Ag_2_Se according to Equation (3):2Ag + Se → Ag_2_Se(3)

Finally, the Ag_2_Se NWs are dispersed in the PEDOT:PSS to fabricate the Ag_2_Se/PEDOT:PSS composite films.

The wire-like structure of the prepared Ag_2_Se NWs is characterized by the FE-SEM image in Figure 1a, while the low magnification FE-SEM image shows a large number of randomly dispersed wire-like nanostructures. The obtained Ag_2_Se wires are seen to be predominantly cylindrical in shape and relatively uniform in size, with lengths of ~1 μm diameters of ~30 nm. In addition, the EDS mapping of Ag and Se in Figure 1b,c indicates the presence of Ag and Se.

The synthesis of the Ag_2_Se NWs are also demonstrated by the XPS analysis, with the Ag and Se signals being evident in the wide-scan spectrum (Figure 2a), and the Ag 3d and Se 3d binding energies being indicated in Figure 2b,c. Thus, the Ag 3d spectrum displays signals at ~368.4 and ~374.4 eV are corresponds with Ag 3_5/2_ and Ag3_3/2_ binding energies, while the two peaks in the Se spectrum at ~54.7 eV and ~53.8 eV correspond to the Se 3d_3/2_ and Se 3d_5/2_, respectively. These results are in agreement with previously reported data for the Ag and Se peaks in Ag_2_Se nanostructures [28].

These XRD data of the synthesized Ag_2_Se NWs were obtained to confirm the composite’s crystal structure, as shown in Figure 3a. These peaks indicate the orthorhombic crystalline phase of the Ag_2_Se product and are in good agreement with the literature values (JCPDS No.24-1041); therefore, they represent the formation of pristine Ag_2_Se (a = 0.705 nm, b = 0.782 nm, and c = 0.434 nm) [29].

The FE-TEM images in Figure 3b,c provide further insight into the details of the Ag_2_Se NWs’ microstructure, including the morphology, crystallinity, and size. Thus, the FE-TEM images in Figure 3b confirm the nanostructures and ~30 nm-diameters of the NWs, in agreement with the FE-SEM images. Moreover, the high-magnification FE-TEM image in Figure 3c reveals the crystalline structure of the Ag_2_Se NWs with two series of lattice planes being indexed, namely the (002) and (200) planes with an interplanar spacing of 0.25 nm and 0.35 nm [30].

Various proportions of the Ag_2_Se were added to PEDOT:PSS solution, along with 5 vol.% DMSO, to generate the Ag_2_Se NW/PEDOT:PSS flexible composite films, which are characterized by the XRD patterns in Appendix A. Here, the Ag_2_Se peaks are seen to systemically increase in intensity with increasing content of Ag_2_Se in the composites, thus confirming the good dispersion of Ag_2_Se and PEDOT:PSS within the composite film.

The ~18 mm black upper surface of the flexible Ag_2_Se NW/PEDOT:PSS composite film with 80 wt.% Ag_2_Se NWs is revealed by the photographic image in Figure 4a. The bendable and resilient nature of thin film indicate its suitability for use in flexible and portable TE modules (Figure 4b). In addition, the cross sectional FE-SEM images of the composite film in Figure 4d reveal the curved morphology, while Figure 4c reveals the even collection of the Ag_2_Se NWs in the polymer matrix.

The room temperature thermoelectric properties of the Ag_2_Se NW/PEDOT:PSS composite films with various Ag_2_Se contents (10, 20, 30, 50, 70 and 80 wt.%) were examined. The *κ* of the pristine Ag_2_Se and the PEDOT:PSS were found to be 0.53 W/m·K and 0.22 W/m·K, which are similar with previously reported values [24,31]. For *S* and *σ* in the simple binary system, the values can be expressed in series- and parallel-connection models [32,33]. Figure 5a,b show the investigated *σ* and *S* values of composite films for the series-connected model (Equation (4)) and the parallel-connected model (Equation (5)):(4)$$S_{c,s}\;=\;\frac{x_{s}\frac{S_{s}}{\kappa_{s}}+\left(1-x_{s}\right)\frac{S_{p}}{\kappa_{p}}}{\frac{x_{s}}{\kappa_{s}}+\frac{\left(1-x_{s}\right)}{\kappa_{p}}}$$
(5)$$S_{c,p}\;=\;\frac{x_{s}\sigma_{s}S_{s}+\left(1-x_{s}\right)\sigma_{p}S_{p}}{x_{s}\sigma_{s}+\left(1-x_{s}\right)\sigma_{p}}$$
where *S_c,s_*, *S_c,p_*, *S_s_*, and *S_p_* are the respective Seebeck coefficients for the series-connected composite, the parallel-connected composite, the Ag_2_Se NWs, and the PEDOT:PSS matrix; *x_s_* is the volume fraction of Ag_2_Se NWs; and *κ_s_*, *κ_p_*, are the respective thermal conductivities of the Ag_2_Se NWs and PEDOT:PSS. The two-component model may also be fitted to *σ* values that vary with the NW loading fraction, and may be specified by Equations (6) and (7):(6)$$\sigma_{c,\;s}{}^{-1}=\;x_{s}\sigma_{s}{}^{-1}+\left(1-x_{s}\right)\sigma_{p}{}^{-1}$$
(7)$$\sigma_{c,p}=\;x_{s}\sigma_{s}+\left(1-x_{s}\right)\sigma_{p}$$
where *σ_c,s_*, *σ_c,p_*, *σ_s_*, and *σ_p_* are the electrical conductivities of the series-connected composite, the parallel-connected composite, the Ag_2_Se NWs, and the PEDOT:PSS matrix, respectively. The values used in the above equation are listed in Appendix A, and the resulting curve using them is shown in Figure 5.

In good agreement with previous studies [25], the Seebeck coefficient of the pristine PEDOT:PSS is seen to be ~12.85 μV/K and is positive, which indicates that PEDOT:PSS exhibited P-type semiconductor behavior. At Ag_2_Se contents above 50 wt.%, however, the Seebeck coefficient becomes negative, thus indicating the N-type character of the composite. Due to the different Seebeck coefficients of the incorporated materials, the Seebeck coefficient of the composite film increases continuously with increasing Ag_2_Se NW content, and displays its maximum value of ~−51.98 μV/K at an Ag_2_Se content of 80 wt.%.

As shown in Figure 5a, the room temperature electrical conductivity of the Ag_2_Se NW/PEDOT:PSS composite film decreases with increasing Ag_2_Se NW contents. This is due to the relatively low electrical conductivity of Ag_2_Se compared to that of the PEDOT:PSS with 5 vol.% DMSO. The thermoelectric power factors (*PF = S^2^·σ*) of the Ag_2_Se NW/PEDOT:PSS composite films are presented in Figure 5c. Due to the combination of increasing *S* and decreasing *σ*, the PF reaches a maximum of ~178.59 μW/m·K^2^ at room temperature for the composite with 80 wt.% Ag_2_Se NW.

The *σ, S,* and power factor of the Ag_2_Se NW/PEDOT:PSS composite films were further analyzed to demonstrate their potential for flexible thermoelectric devices. For example, the influence of repeat bending cycles upon the thermoelectric properties of the Ag_2_Se NW/PEDOT:PSS composite film with 70 and 80 wt.% Ag_2_Se NW is shown in Figure 6. As the number of bending cycles increases, the durability of the film and, hence, the thermoelectric properties decrease. Nevertheless, the degree of decline in thermoelectric properties is not significant. From these results, it can be seen that the Ag_2_Se NW/PEDOT:PSS composite film with 70 and 80 wt.% Ag_2_Se NW maintains excellent durability after 1000 repeat cycles.

Figure 7 illustrates the *σ, S,* and *PF* of composite film with 80 wt.% Ag_2_Se NWs under different temperatures. The Seebeck coefficient is seen to gradually decrease with increasing temperature, while the electrical conductivity initially increases until a temperature of ~400 K is reached, after that, it subsequently decreases with further increase in temperature. This trend is similar to that observed in previous studies using Ag_2_Se semiconducting materials, and these trends are predicted as a result of changes in carried concentration [34,35]. The temperature-dependence of *S* and *σ* within the composite can be understood by the relationship with the carrier concentration:*σ* = *n* · *e* · *μ*(8)
(9)$$S=\;\frac{8\cdot \pi^{2}\cdot k_{B}{}^{2}}{3\cdot e\cdot h^{2}}\cdot m^{*}\cdot T\cdot {\left(\frac{\pi }{3\cdot n}\right)}^{\frac{2}{3}}$$
where *n*, *e*, *μ*, *k_B_, h,* and *m** are the carrier concentration, electron charge, carrier mobility, Boltzmann constant, Planck constant, and effective mass of the carrier, respectively. In the present work, the maximum *PF* of the Ag_2_Se NW/PEDOT:PSS composite film is found to be ~183.29 μW/m·K^2^ at 400 K. In addition, Appendix A indicates that the temperature-dependent thermoelectric properties of the synthesized composite film with 80 wt.% Ag_2_Se NW display no significant difference over multiple heating and cooling cycles.

## 4. Conclusions

In this report, Ag_2_Se NW/PEDOT:PSS composite films with different Ag_2_Se NW contents were prepared via a simple method. The as-prepared Se NWs were used to fabricate Ag_2_Se NWs. The synthesized Ag_2_Se NWs exhibited a wire-like 1D structure with diameters of ~30 nm and lengths of ~1 μm. The PEDOT:PSS used in this study was treated with DMSO for enhanced electrical conductivity. The synthesized NWs were dispersed in the DMSO-treated PEDOT:PSS solution and sonicated to obtain a uniform distribution. FE-SEM, FE-TEM, EDS, XRD, and XPS analyses were used to verify the successful synthesis and micromorphology of the composite film. The Seebeck coefficient of the Ag_2_Se NW/PEDOT:PSS composite films displayed an increasing trend with increasing Ag_2_Se NW contents due to the relatively high Seebeck coefficient of Ag_2_Se. Consequently, the composite film with 80 wt.% Ag_2_Se NW showed the highest power factor of 183.29 μW/m·K^2^ at 400 K, whose *PF* value is ~14 times higher than that of the pure PEDOT:PSS film. The composite also showed outstanding durability after 1000 repeat bending cycles. Hence, it was concluded that composites manufactured using Ag_2_Se NWs and PEDOT:PSS could be promising materials for the production of high-performance flexible thermoelectric devices.

## Figures and Tables

**Figure 1 polymers-12-02932-f001:**
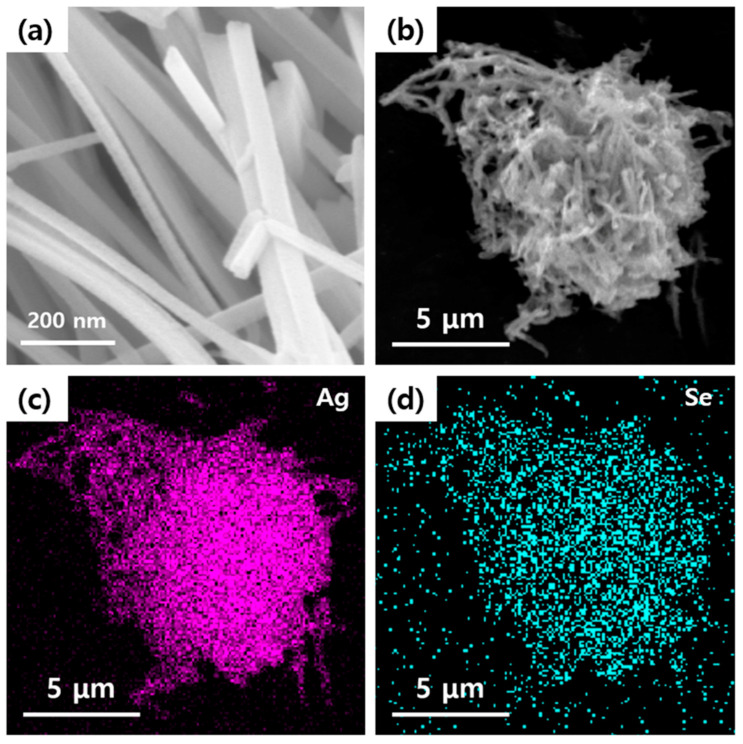
(**a**,**b**) FE-SEM images and the corresponding EDS elemental mapping of (**c**) Ag and (**d**) Se atoms.

**Figure 2 polymers-12-02932-f002:**
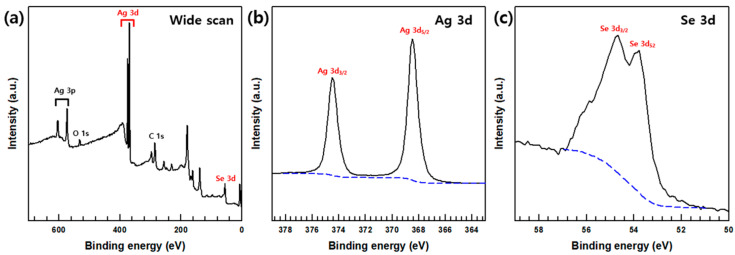
(**a**) XPS survey spectrum of Ag_2_Se NWs, high-resolution XPS spectra of (**b**) Ag 3d, and (**c**) Se 3d core level.

**Figure 3 polymers-12-02932-f003:**
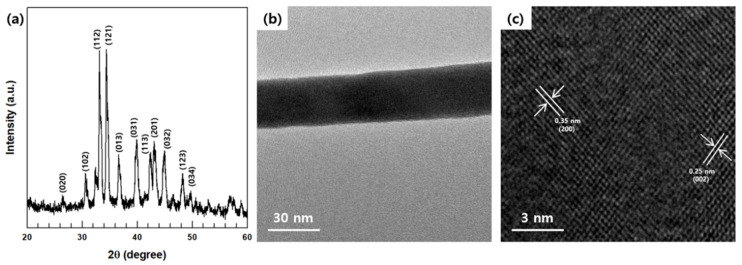
(**a**) XRD patterns, (**b**) Low and (**c**) high magnification images of synthesized Ag_2_Se NWs.

**Figure 4 polymers-12-02932-f004:**
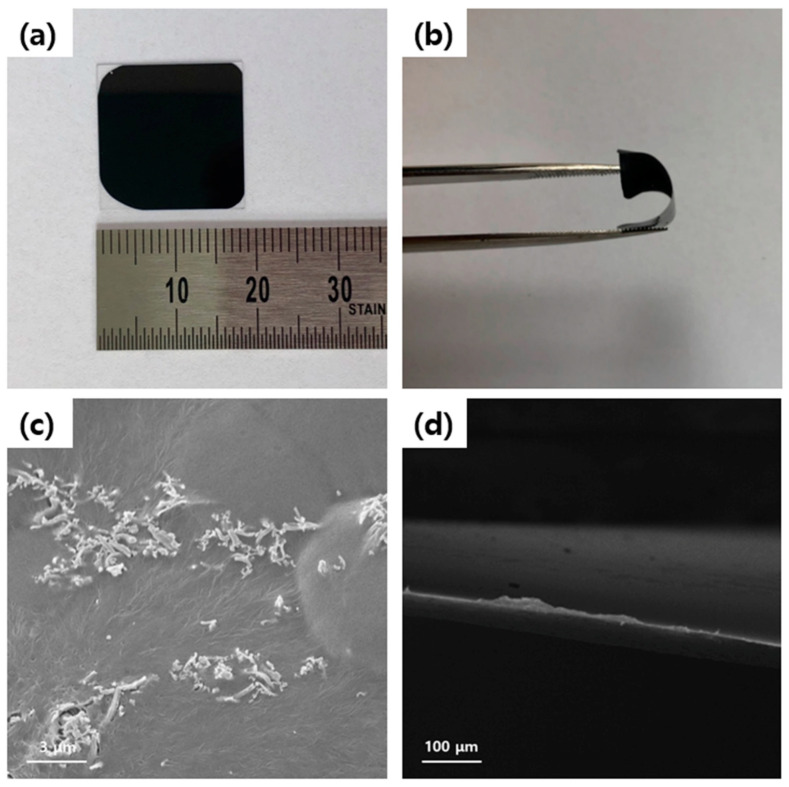
(**a**,**b**). Digital photo image of Ag_2_Se NW/PEDOT:PSS composite films. (**c**) Surface and (**d**) cross-sectional FE-SEM images of Ag_2_Se NW/PEDOT:PSS composite films.

**Figure 5 polymers-12-02932-f005:**
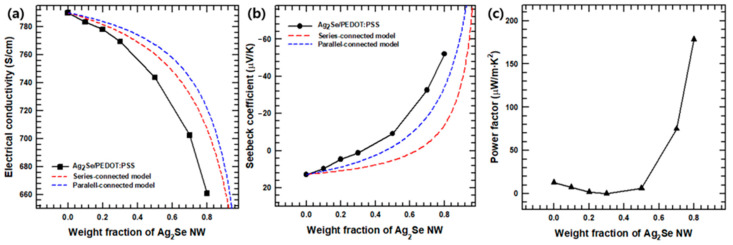
(**a**) Electrical conductivity, (**b**) Seebeck coefficient, and (**c**) power factor values of the Ag_2_Se NW/PEDOT:PSS composite films with two models (series and parallel connected models) as a function of the Ag_2_Se NW contents.

**Figure 6 polymers-12-02932-f006:**
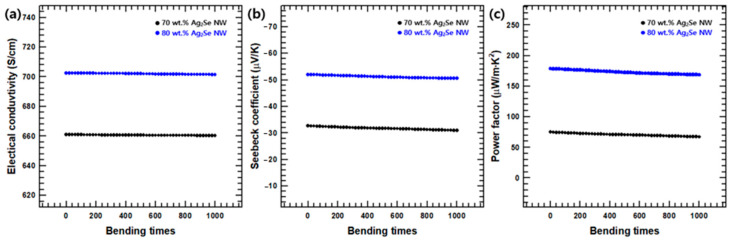
(**a**) Electrical conductivity, (**b**) Seebeck coefficient, and (**c**) power factor of Ag_2_Se NW/PEDOT:PSS composite films with 70 and 80 wt.% of Ag_2_Se NW as a function of bending cycles.

**Figure 7 polymers-12-02932-f007:**
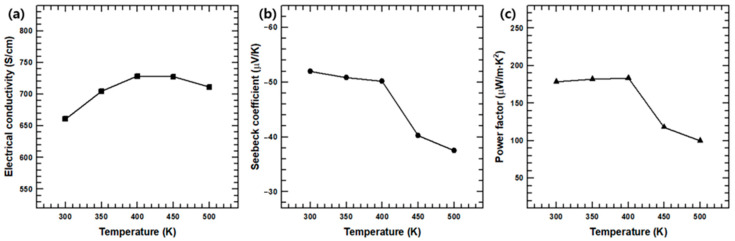
(**a**) Electrical conductivity, (**b**) Seebeck coefficient, and (**c**) power factor values of Ag_2_Se NW/PEDOT:PSS composite films with 80 wt.% of Ag_2_Se as a function of temperature.

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
