# Peer review of "Fabrication of PEDOT:PSS/Ag2Se Nanowires for Polymer-Based Thermoelectric Applications"

_polymers, 2020, doi:10.3390/polym12122932_

Round 1
Reviewer 1 Report
In this paper Ag2Se NW/PEDOT:PSS composite films with different Ag2Se NW contents were prepared via a simple original method. Composite film with 80 wt.% Ag2Se showed the highest power factor and outstanding durability after repeat bending cycles. Thus, the study is of interest in my judgment and it can be recommended for publication after a minor revision:
1. Figure 3 (a) depicts XRD patterns of Ag2Se obtained for NW -powder. However, it may be of interest to compare XRD patterns of Ag2Se NW -powder and Ag2Se composite films, because the nanowire shape of Ag2Se crystals can lead to the formation of texture and, hence, the appearance of anisotropy of properties.
2. Equation (1) contains an error (C6H8O6 on the right side of the equation must be corrected for C6H6O6).
3. On page 8 line 209-212 it says that «The Seebeck coefficient is seen to gradually decrease with increasing temperature, while the electrical conductivity initially increases until a temperature of ~400K is reached, after that, subsequently decreases with further increase in temperature. This trend is similar to that observed in previous studies using Ag2Se semiconducting materials.» Can the authors provide an explanation for this phenomenon? Perhaps the percolation threshold is reached here? Or are there other reasons?
Author Response
Thank you for your sincere comments on this paper.
We have examined each point and revised the manuscript accordingly.
Reviewer’s comments:
Reviewer #1: In this paper Ag2Se NW/PEDOT:PSS composite films with different Ag2Se NW contents were prepared via a simple original method. Composite film with 80 wt.% Ag2Se showed the highest power factor and outstanding durability after repeat bending cycles. Thus, the study is of interest in my judgment and it can be recommended for publication after a minor revision:
- Figure 3 (a) depicts XRD patterns of Ag2Se obtained for NW -powder. However, it may be of interest to compare XRD patterns of Ag2Se NW -powder and Ag2Se composite films, because the nanowire shape of Ag2Se crystals can lead to the formation of texture and, hence, the appearance of anisotropy of properties.
Fig. 1. (a) XRD patterns of pristine Ag2Se NW and Ag2Se NW/PEDOT:PSS composite film. (b) Low- and (c) high magnification of surface FE-SEM image of Ag2Se NW/PEDOT:PSS composite films.
After receiving reviewer’s comment, the XRD patterns of pristine nanowires and composite films is analyzed. No significant difference was found in two peaks. The surface FE-SEM images of composite films are shown in Figs 1a. and 1b. In this FE-SEM images, it was confirmed that the NW form did not collapse much. Therefore, we can confirm that the nanostructure of Ag2Se NWs did not significantly changed during the drop-casting process.
- Equation (1) contains an error (C6H8O6 on the right side of the equation must be corrected for C6H6O6).
After reviewer’s comment, we revise the contents. Thank you for your comment.
- On page 8 line 209-212 it says that «The Seebeck coefficient is seen to gradually decrease with increasing temperature, while the electrical conductivity initially increases until a temperature of ~400K is reached, after that, subsequently decreases with further increase in temperature. This trend is similar to that observed in previous studies using Ag2Se semiconducting materials.» Can the authors provide an explanation for this phenomenon? Perhaps the percolation threshold is reached here? Or are there other reasons?
The change in thermoelectric properties of the composite film according to temperature is similar to that of other thermoelectric materials using Ag2Se [1,2]. In these references, this trend of thermoelectric properties is explained through changes in carrier mobility and carrier concentration. We already know that the relationship with the Seebeck coefficient, electrical conductivity, carrier mobility, and carrier concentration. However, we have studied with another research group for the demonstration of electrical transport. The research results from the electrical transport in the composites will be submitted for the publication of another research paper. In further studies, we will research more and report on this trend.
Additionally, we added information on the relationships at last paragraph at main article.
“The temperature-dependence of S and σ within the composite can be understood by the relationship with the carrier concentration:
σ = n · e · μ (8)
(9)
where n, e, μ, kB, h, and m* are the carrier concentration, electron charge, carrier mobility, Boltzmann constant, Planck constant, and effective mass of the carrier, respectively.”
References
- Ding, Y.; Qiu, Y.; Cai, K.; Yao, Q.; Chen, S.; Chen, L.; He, J., High performance n-type Ag2Se film on nylon membrane for flexible thermoelectric power generator. Nature Commun. 2019, 10 (1), 1-7.
- Jiang, C.; Ding, Y.; Cai, K.; Tong, L.; Lu, Y.; Zhao, W.; Wei, P., Ultrahigh Performance of n-Type Ag2Se Films for Flexible Thermoelectric Power Generators. ACS Appl. Mater. Inter. 2020, 12 (8), 9646-9655.

Reviewer 2 Report
1: The title is very long and can be modified simply as Fabrication of PEDOT:PSS: Ag2Se nanowires for polymer based thermoelectric applications
2: In the abstract the authors should mention the methos of fabrication. E.g ---- were fabricated through----
3: Are the method of synthesis quoted in the experimental section are novel developments by the authors? if not they should provide the relevant literature followed for synthesis procedure and describe the novelty and importance of their work.
3: The author observed an increase in conductivity to a certain limit with increase in temperature and then it decreases. Please mention the reason for the interest of the readers.
4: Please provide pictures of the materials used. Mostly schematic diagrams have been provided, so to the possible extent, real pics should be provided as well alongside schematic diagrams. Moreover, a little comparison on similar work done by the others will be helpful to recognize the difference in their work.
Author Response
Thank you for your sincere comments on this paper.
We have examined each point and revised the manuscript accordingly.
Reviewer’s comments:
Reviewer #2:
1: The title is very long and can be modified simply as Fabrication of PEDOT:PSS: Ag2Se nanowires for polymer based thermoelectric applications.
After receiving reviewer’s comment, we revise the title of the paper. Thank you for your sincere comment.
“Fabrication of PEDOT:PSS/Ag2Se nanowires for polymer based thermoelectric applications.”
2: In the abstract the authors should mention the methods of fabrication. E.g ---- were fabricated through----
After receiving reviewer’s comment, we added the simple methods of fabrication.
“The Ag2Se nanowire are first fabricated with solution mixing. After that, Ag2Se NW/PEDOT:PSS composite film was fabricated using a simple drop-casting method.”
3: Are the method of synthesis quoted in the experimental section are novel developments by the authors? if not they should provide the relevant literature followed for synthesis procedure and describe the novelty and importance of their work.
The Ag2Se NWs used in this study were fabricated with simple solution mixing, as reported in previous studies [1,2]. However, there are no reports about making film using drop casting after dispersing Ag2Se NWs in PEDOT:PSS matrix. Therefore, in the fabrication of flexible Ag2Se NW/PEDOT:PSS composite film and the discussion of thermoelectric enhancement of the composite film is our new idea, which can have important practical applications for the high performance polymer-based thermoelectric devices.
References
- Ding, Y.; Qiu, Y.; Cai, K.; Yao, Q.; Chen, S.; Chen, L.; He, J., High performance n-type Ag2Se film on nylon membrane for flexible thermoelectric power generator. Nature Commun. 2019, 10 (1), 1-7.
- Jiang, C.; Ding, Y.; Cai, K.; Tong, L.; Lu, Y.; Zhao, W.; Wei, P., Ultrahigh Performance of n-Type Ag2Se Films for Flexible Thermoelectric Power Generators. ACS Appl. Mater. Inter. 2020, 12 (8), 9646-9655.
4: The author observed an increase in conductivity to a certain limit with increase in temperature and then it decreases. Please mention the reason for the interest of the readers.
The change in thermoelectric properties of the composite film according to temperature is similar to that of other thermoelectric materials using Ag2Se [1,2]. In these references, this trend of thermoelectric properties is explained through changes in carrier mobility and carrier concentration. We already know that the relationship with the Seebeck coefficient, electrical conductivity, carrier mobility, and carrier concentration. However, we have studied with another research group for the demonstration of electrical transport. The research results from the electrical transport in the composites will be submitted for the publication of another research paper. In further studies, we will research more and report on this trend.
Additionally, we added information on the relationships at last paragraph at main article.
“The temperature-dependence of S and σ within the composite can be understood by the relationship with the carrier concentration:
σ = n · e · μ (8)
(9)
where n, e, μ, kB, h, and m* are the carrier concentration, electron charge, carrier mobility, Boltzmann constant, Planck constant, and effective mass of the carrier, respectively.”
References
- Ding, Y.; Qiu, Y.; Cai, K.; Yao, Q.; Chen, S.; Chen, L.; He, J., High performance n-type Ag2Se film on nylon membrane for flexible thermoelectric power generator. Nature Commun. 2019, 10 (1), 1-7.
- Jiang, C.; Ding, Y.; Cai, K.; Tong, L.; Lu, Y.; Zhao, W.; Wei, P., Ultrahigh Performance of n-Type Ag2Se Films for Flexible Thermoelectric Power Generators. ACS Appl. Mater. Inter. 2020, 12 (8), 9646-9655.
5: Please provide pictures of the materials used. Mostly schematic diagrams have been provided, so to the possible extent, real pics should be provided as well alongside schematic diagrams. Moreover, a little comparison on similar work done by the others will be helpful to recognize the difference in their work.
Fig. 2. Overall scheme on real pictures of materials used in this study. (a) Selenium (IV) oxide (SeO2, 99%). (b) synthesized Se NWs. (c) Synthesized Ag2Se NWs. (d) Ag2Se NW/PEDOT:PSS composite film.
At figure, the overall scheme of preparation of the Ag2Se NW/PEDOT:PSS composites film is shown. Se NWs were first synthesize with the reduction of SeO2. After that, Ag2Se NWs were synthesized with Ag precursor and synthesized Se NWs. Finally, the Ag2Se NWs and PEDOT:PSS is mixed and drop-casted to fabricate the composite film.
Some digital photographs are also shown in this figure. Figure (a) shows pictures of purchased reagents. At Figure (b) and (c), the synthesized Se and Ag2Se NWs are shown, and the color of Se and Ag2Se NWs are dark red and black. The photograph of black surface of the composite film is shown in Figure (d)
We hope that this revision addresses the reviewer’s comments.
Sincerely
